# A Rapid and Sensitive Liquid Chromatography-Tandem Mass Spectrometry Bioanalytical Method for the Quantification of Encorafenib and Binimetinib as a First-Line Treatment for Advanced (Unresectable or Metastatic) Melanoma—Application to a Pharmacokinetic Study

**DOI:** 10.3390/molecules28010079

**Published:** 2022-12-22

**Authors:** Mohamed M. Hefnawy, Mohammed M. Alanazi, Abdullah M. Al-Hossaini, Abdulaziz I. Alnasser, Adel S. El-Azab, Yousef A. Bin Jardan, Mohamed W. Attwa, Manal A. El-Gendy

**Affiliations:** 1Department of Pharmaceutical Chemistry, College of Pharmacy, King Saud University, Riyadh 11451, Saudi Arabia; 2Department of Analytical Chemistry, Faculty of Pharmacy, Mansoura University, Mansoura 35516, Egypt; 3Department of Pharmaceutics, College of Pharmacy, King Saud University, Riyadh 11451, Saudi Arabia

**Keywords:** LC–MS/MS, melanoma, encorafenib, binimetinib, rat plasma, pharmacokinetics

## Abstract

The combination regimen targeting BRAF and MEK inhibition, for instance, encorafenib (Braftovi^™^, ENF) plus binimetinib (Mektovi^®^, BNB), are now recommended as first-line treatment in patients with unresectable or metastatic melanoma with a BRAF V600-activating mutation. Patients treated with combination therapy of ENF and BNB demonstrated a delay in resistance development, increases in antitumor activity, and attenuation of toxicities compared with the activity of either agent alone. However, the pharmacokinetic profile of the FDA-approved ENF and BNB is still unclear. In this study, a rapid and sensitive LC-MS/MS bioanalytical method for simultaneous quantification of ENF and BNB in rat plasma was developed and validated. Chromatography was performed on an Agilent Eclipse plus C18 column (50 mm × 2.1 mm, 1.8 µm), with an isocratic mobile phase composed of 0.1% formic acid in water/acetonitrile (67:33, *v*/*v*, pH 3.2) at a flow rate of 0.35 mL/min. A positive multiple reaction monitoring (MRM) mode was chosen for detection and the process of analysis was run for 2 min. Plasma samples were pre-treated using protein precipitation with acetonitrile containing spebrutinib as the internal standard (IS). Method validation was assessed as per the FDA guidelines for the determination of ENF and BNB over concentration ranges of 0.5–3000 ng/mL (r^2^ ≥ 0.997) for each drug (plasma). The lower limits of detection (LLOD) for both drugs were 0.2 ng/mL. The mean relative standard deviation (RSD) of the results for accuracy and precision was ≤ 7.52%, and the overall recoveries of ENF and BNB from rat plasma were in the range of 92.88–102.28%. The newly developed approach is the first LC–MS/MS bioanalytical method that can perform simultaneous quantification of ENF and BNB in rat plasma and its application to a pharmacokinetic study. The mean result for C_max_ for BNB and ENF was found to be 3.43 ± 0.46 and 16.42 ± 1.47 µg/mL achieved at 1.0 h for both drugs, respectively. The AUC_0-∞_ for BNB and ENF was found to be 18.16 ± 1.31 and 36.52 ± 3.92 µg/mL.h, respectively. On the other hand, the elimination half-life (t_1/2kel_) parameters for BNB and ENF in the rat plasma were found to be 3.39 ± 0.43 h and 2.48 ± 0.24 h, and these results are consistent with previously reported values.

## 1. Introduction

Incidence rates of skin melanoma have risen in recent decades, now representing 1% of all skin cancers. In 2022, an estimated 108,480 new cases were diagnosed, and 11,990 persons are expected to die from melanoma [1]. Approximately 50% of patient’s detected with metastatic melanoma have a protein kinase B-Raf (BRAF) point mutation. Mutations at codon 600 of the BRAF gene lead to tumor proliferation through increased signal transduction of the mitogen-activated protein kinase (MAPK) pathway. BRAF V600E is the most common V600 point mutation, occurring in 84.6% of BRAF-mutated melanomas [2,3]. The development of active molecular agents and immune suppression inhibitors has advanced the treatment of metastatic melanoma [4]. The first metastatic melanoma combination therapy consisted of the use of the BRAF vemafenib and dabrafenib inhibitors and the MEK cobimetinib and trametinib inhibitors, which have shown efficacy in the treatment of BRAFV600 mutation-positive or metastatic melanoma patients. However, even with these combination therapies, resistance still remains a significant problem; 80% in the first three years of therapy are immune [4].

Encorafenib plus binimetinib was approved in 2018 for the treatment of patients with unresectable or metastatic melanoma with BRAF V600E or V600K mutations [5]. The combination of the BRAF inhibitor ENF plus the MEK inhibitor BNB gives a suitable outcome over other BRAF/MEK combinations [6]. ENF and BNB are inhibitors of protein kinases in the MAPK pathway. ENF targets *BRAF* V600E, V600D, and V600K mutant kinases. BNB is a reversible inhibitor of MEK1 and MEK2. The inhibition of BRAF and MEK kinases results in the inhibition of extracellular signal-regulated kinase (ERK) phosphorylation, which ultimately leads to decreased cell proliferation. The combination of BRAF and MEK inhibition decreases resistance, increases antitumor activity, and attenuates toxicities compared with the activity of either agent alone [7,8]. 

Few LC–MS/MS methods have been reported for the quantification of BNB in biological matrices either alone [9] or in combination with other anticancer drugs in biological fluids [10,11,12]. Recently, a metabolic stability study has been applied for ENF and BNB in the human liver microsome matrix by utilizing the LC–MS/MS technique [13]. An extensive literature review revealed that reports describing an analytical method for simultaneous quantification of ENF and BNB in biological fluids with the application to pharmacokinetic study are not described. The objective of the present study was the determination of ENF and BNB in rat plasma with a short analysis time (2 min) and the development of a sensitive and specific LC–MS/MS method with the application to pharmacokinetic studies. The newly validated assay has a wide linear range, lower sensitivity (0.2 ng/mL) and employs a lower plasma volume (50 µL) for processing than other bioanalytical methods [11,13]. As far as we know, this newly developed approach is the first study applied with desired accuracy and precision for monitoring the pharmacokinetic behavior of ENF and BNB in rats, and parameters such as C_max_, T_max_, t_1/2kel_, AUC_0–24_, and AUC_0–∞_ were evaluated.

## 2. Results and Discussion

### 2.1. Optimization of Chromatographic Conditions and MS Detections 

The present study aimed to develop and validate a fast and sensitive method to quantitatively determine ENF and BNB in rat plasma. Chromatographic conditions, such as the nature of the mobile phase and its composition, were optimized through many trials in order to obtain the best resolution and the highest signal for ENF and BNB and spebrutinib (IS). The pH of the aqueous mobile phase, 0.1% formic acid solution, was adjusted to 3.2, as higher pH values led to peak tailing and long elution time. Several mobile phase compositions of ammonium formate buffer, ammonium acetate buffer, 0.1% formic acid, 0.1% trifluoracetic acid and 0.1% acetic acid in water, with either acetonitrile or methanol, were tested in an isocratic mode regarding peak shape, response, analysis time and peak area. Furthermore, selected mobile phases were examined with different ratios of acetonitrile percentage (20–90%) and water, each mixed with 0.1% formic acid. The percentage of acetonitrile in the mobile phase had a significant effect on the separation and retention time of ENF and BNB and IS. An increasing percentage of acetonitrile resulted in overlapped peaks and a poor separation, while decreasing acetonitrile percentage resulted in a long running time. The optimized mobile phase was composed of 0.1% formic acid in water (67%), and 33% acetonitrile was shown to improve the signal-to-noise ratio and thus found to be suitable for the chromatographic separation of the analytes at a flow rate of 0.35 mL/min. Different stationary phases were tried for chromatographic separation, polar and non-polar ones, with a different column pack of either cyano-, phenyl- or octyl (C8) and octadecyl (C18), with different dimensions. However, good results were achieved using Eclipse plus C18 column (50 mm × 2.1 mm i.d., 1.8 µm; Agilent Technologies Palo Alto, CA, USA). In addition, we investigated the use of different internal standards, such as repaglinide, nateglinide, pemigatinib, chloroquine, and hydroxychloroquine, but such internal standards either gave poor peaks or led to overlapping with BNB or ENF. A chemically similar IS, spebrutinib, was chosen as the method’s IS, whereas it has a closer extraction recovery and performance characteristics to ENF and BNB. Sample processing by liquid–liquid extraction and protein precipitation using different solvents was tried. It was found that protein precipitation utilizing acetonitrile is the optimum method with regard to simplicity, affordability and easier sample processing. Chromatographic separation of ENF, BNB, and IS was achieved with good resolution over a run time of 2.0 min (Figure 1). 

For the highest intensity of the protonated molecular ions, different MS/MS parameters, such as the desolvation and the nebulizer gases, were adjusted to achieve a better spray shape without affecting the sensitivity of ENF, BNB and IS (Table 1). The ENF, BNB and IS were found to have a higher response in positive ion mode with low noise levels. Therefore, the positive ion mode for ENF, BNB and IS was selected, yielding high-abundance fragment ions of: *m/z* 540.1→359.1 for ENF, 441.0→165.0 for BNB and 424.1 → 370.1 for IS, respectively, as shown in Figure 2.

### 2.2. In-Study Validation

The proposed LC-MS/MS method was fully validated, guided by the United States FDA guideline for the validation of bioanalytical methods [14]. The studied validation parameters in the rat plasma involved determining method linearity and range, selectivity, precision and accuracy, extraction recovery, carry-over, dilution integrity, matrix effect and stability. A linear range of the developed assay was established over a wide concentration range 0.5–3000 ng/mL in rat plasma. The linear regression of ENF and BNB attained during the method validation is listed in Table 2. The regression equations achieved by least squared regression for ENF and BNB were; y = 0.0012x + 0.0654, r^2^ = 0.999, and y = 0.0023x + 0.0431, r^2^ = 0.998; for ENF and BNB, respectively, where y is the peak area ratio of D/IS and x is the concentration (ng/mL). The results confirmed the linearity and reproducibility of the assay method. The LLOD of ENF and BNB in rat plasma was 0.2 ng/mL, confirming the applicability of the developed assay for the quantification of trace concentrations ENF and BNB in plasma.

Representative total ion chromatograms of ENF and BNB and IS in rat plasma are demonstrated in Figure 3, which indicated that the analysis of blank plasma samples and plasma spiked with lower limit quality control (LLOQ), lower quality control (LQC), middle quality control (MQC), and high quality control (HQC) levels showed that there were no interferences at the retention times of ENF and BNB and IS, confirming the selectivity of the method. The carry-over in the blank sample was less than 20% of LLOQ for ENF and BNB and less than 5% of the response for IS after injection of the upper limit of quantification (ULOQ) of the calibration curve [14].

Four concentrations of QC samples (LLOQ, LQC, MQC, HQC) in six replicates were used to check the intra- and inter-assay precision and accuracy. The accuracy and precision results of ENF and BNB determination are summarized in Table 3. The values for intra-day and inter-day precision and accuracy were 0.33–6.23% and 92.88–102.28% for ENF and 0.38–7.52% and 94.00–101.31% for BNB, respectively; these values met the acceptance criteria of the guidelines; LLOQ within 20% and the other QCs within 15% [14]. 

The mean percent recoveries following the sample preparation of ENF and BNB from the plasma matrix were examined at three QC levels (1.5, 1800, 2400 ng/mL) in six replicates were 94.18% and 93.41%, respectively. Moreover, the mean % recovery of IS was not less than 95.28 ± 1.74 for all tested samples presented in Table 4.

The matrix effect (ME) for ENF, BNB and IS was calculated as low and high QC samples by dividing the peak area in the presence of matrix components by the peak area in the neat standard solution of the analyte. The IS normalized ME is calculated by dividing the ME of the analyte by the ME of the IS. The RSD of IS-normalized ME of the six batches of the plasma was less than 15%. For BNB, it was 1.70 and 0.27 for LQC and HQC, respectively. For ENF, it was 1.38 and 0.91 for LQC and HQC, respectively, indicating that ion suppression/enhancement from the plasma was insignificant.

Six replicates of plasma samples spiked with high concentrations of each drug beyond the linear range were processed and analyzed using a dilution factor of two and four in order to examine the accuracy of the method after dilution. The results were within the method quantitation range with RSD within 1.08–1.08%, and accuracy results varied from 94.66 to 99.26% (Table 5). This approves the minimal effect of dilution on the outcomes of the developed assay.

An important process of bioanalytical method validation is stability assessment. Stability of BNB and ENF was studied throughout the analysis of three QC samples (LQC, MQC and HQC) of each drug after the application of the different storage conditions. The different parameters investigated include short-term stability at room temperature for 24 h, autosampler stability at 10 °C for 24 h, three freeze and thaw cycles after storing at −80 °C, and long-term stability at −80 °C for 30 days. The results of stability experiments were satisfactory and complied with the accuracy criteria of ±15% of its theoretical concentration. Table 6 shows the detailed results.

### 2.3. Application to the Pharmacokinetic Study 

The newly developed and validated LC-MS/MS assay was effectively applied to evaluate BNB and ENF in rat plasma for a pharmacokinetic study after oral administration of 3.8 mg/kg BNB and 20 mg/kg ENF for four healthy male Wistar rats under fasting conditions. The assay specificity and sensitivity considered to be adequate for precisely characterizing the plasma pharmacokinetic parameters for the BNB and ENF are demonstrated in Table 7. The mean plasma concentration-time profile is displayed in Figure 4. The mean result for C_max_ for BNB and ENF was found to be 3.43 ± 0.46 and 16.42 ± 1.47 µg/mL achieved at 1.0 h for both drugs, respectively. The AUC_0-∞_ for BNB and ENF was found to be 18.16 ± 1.31 and 36.52 ± 3.92 µg/mL·h, respectively. The results achieved were found to be in close accord with previously reported values [15,16]. The values acquired in the current investigation for BNB for t_1/2kel_ and Cl/F parameters are consistent with in vivo BNB PK studies [15]. Furthermore, the T_max_ parameter for BNB is similar to that represented in a recently published paper [11]. On the other hand, the elimination half-life (t_1/2kel_) parameters for BNB and ENF in the rat plasma were found to be 3.39 ± 0.43 and 2.48 ± 0.24 h, and these results are consistent with previously reported values [16,17].

## 3. Experimental

### 3.1. Chemicals and Reagents

Reference standards of encorafenib (99.0%), binimetinib (99.0%), and spebrutinib (internal standard, IS, 97.6%) were purchased from Med Chem Express (Monmouth Junction, NJ, USA). HPLC-grade acetonitrile and formic acid dimethyl sulfoxide (DMSO) were obtained from Sigma-Aldrich (West Chester, PA, USA). Ultrapure water was prepared by an in-house Milli-Q Millipore Water System (Millipore, Billerica, MA, USA). All other solvents and reagents used were of analytical grade. Drug-free rat plasma was obtained from the Animal Care Centre (College of Pharmacy, King Saud University, Saudi Arabia). Rat plasma was used in this study instead of human plasma because there was a significant correlation between the lipoprotein lipid and protein profiles in human and rat plasma [18].

### 3.2. LC-MS/MS Conditions

An Acquity water UPLC (model code (UPA) and serial number (A11UPA448M)) was used for chromatographic separation, while Acquity TQD MS (model code (TQD) and serial number (QBB1203)) was used for mass analysis of eluted analytes peaks. Samples were separated on a reversed-phase Acquity^®^ UPLC BEH C18 column (1.7 µm particle size, 50 mm × 2.1 mm ID) in isocratic mode. The mobile phase was composed of a mixture of water containing 0.1% formic acid and acetonitrile (67:33, *v*/*v*, pH 3.2) at a flow rate of 0.35 mL/min. The column temperature and autosampler were kept constant at a room temperature of 25 °C. The injection volume was 5.0 µL, and the total run time was 2 min. The solvents were filtered through membrane filters (0.22 µm) obtained from Chrom Tech (Kent, UK). The needle was washed after each injection with a mixture of methanol and water (80:20). Mass spectrometry parameters for the triple quadrupole mass analyzer (TQD MS) were optimized to attain a good separation of ENF, BNB and spebrutinib (SPB: internal standard, IS) with good sensitivity. The ENF, BNB, and IS were estimated using TQD MS that was operated in positive mode (ESI+). The tuning parameters for ENF, BNB, and IS were chosen using IntelliStart^®^ software that was readjusted manually in combined mode (fluidics and LC) to enhance chromatographic peak parameters such as signal intensity and selectivity. Nitrogen (650 L/h) was used as drying gas at 350 °C. The cone gas flow rate was kept at 100L/H. Argon (0.14 mL/min) was used as a collision gas inside the fragmentation cell. The cone voltages for BNB, ENF and SPB were set at 44, 54 and 58 (V), respectively. Extractor voltage, capillary voltage, and RF lens were set at 3.0 (V), 4 (kV), and 0.1 (V), respectively. A multiple reaction monitoring (MRM) mass analyzer in positive ion mode was utilized for the detection of ENF, BNB, and IS to avoid interference from rat plasma matrix constituents and to increase the selectivity and sensitivity of the developed method.

### 3.3. Preparation of Stock, Standard, Calibrators and Quality Control Samples

Primary stock solutions of ENF, BNB and SPB (IS) were prepared separately in dimethyl sulfoxide (DMSO) at a concentration of 1.0 mg/mL and stored at −20 °C. Successive working solutions of ENF and BNB were additionally obtained through dilution using ultrapure water at concentrations of 0.05, 0.5, 5 and 20 µg/mL. A working solution of IS was prepared in ultrapure water at a concentration of 2 µg/mL. Calibrators at concentrations of 0.5,1, 5, 50, 100, 200, 500, 1000, 2600 and 3000 ng/mL for ENF and BNB were prepared in blank rat plasma from the intermediate solutions. Different quality control samples at 0.5 ng/mL for the LLOQ, 1.5 ng/mL for the QC sample at low concentration, 1800 ng/mL for the QC sample at mid concentration, and 2400 ng/mL for the QC sample at high concentration were prepared by spiking the appropriate volume of the intermediate solutions with blank rat plasma. The peak area ratios of each drug to IS were treated to obtain the calibration curve of each drug. Alternatively, the corresponding regression equation was derived.

### 3.4. Sample Preparation

Frozen plasma samples were thawed before analysis at room temperature. A volume of 50 µL of working IS solution (100 ng/mL) was added to the 50 µL plasma in 2.0 mL disposable polypropylene micro centrifuge tubes. Each tube was diluted to 750 µL with ultrapure water and vortex for 30 s. A total of 500 µL of acetonitrile was added to the spiked plasma samples to precipitate the plasma proteins and mixed for 60 s. The tubes were subsequently vortexed for 60 s and centrifuged at 10,000 rpm at 5 °C for 12 min. An aliquot of 300 μL of the upper apparent solution was mixed with 700 μL ultrapure water, vortex mixed, and a 100 μL was transferred into a vial for analysis. Finally, a volume of 5 μL was injected into the LC-MS/MS system using an autosampler.

### 3.5. Pre-Study Validation

Intensive validation studies for analyzing ENF and BNB in rat plasma were performed following the US-FDA guidelines [14]. The studied validation parameters in the rat plasma involved determining method selectivity, linearity and range, precision and accuracy, extraction recovery, carry-over, dilution integrity, matrix effect and stability. Method selectivity was performed by estimating the interference from endogenous components at the retention time of ENF, BNB, and IS in blank rat plasma from six different lots. The responses of less than 20% of the LLOQ for ENF, BNB and <5% of the IS were accepted [14]. 

The calibration curves were evaluated in rat plasma by plotting the active response for each linearity solution against their respective theoretical concentrations. The concentrations used for ENF and BNB were 0.5,1, 5, 50, 100, 200, 500, 1000, 2600 and 3000 ng/mL. We used the least squares statistical method to compute the calibration curve equations (*y* = *mx*+ *b*). The linear fit was confirmed utilizing the coefficient of determination (r^2^) value, which showed linearity in the range of 0.5 to 3000 ng/mL. The deviation of non-zero calibrators should be ± 15% of the nominal values except at LLOQ where the calibrators should be ± 20% of the nominal concentrations [14]. 

Intra-day and inter-day accuracy and precision were estimated by analyzing a calibration curve (in triplicate) and spiked plasma samples at the lower limit (LLOQ) in addition to three different QC levels (LQC, MQC, and HQC, respectively) in six-fold on three different days. The examined levels were 0.5 ng/mL (LLOQ), 1.5 ng/mL (LQC), 1800.0 ng/mL (MQC) and 2400.0 ng/mL (HQC) for BNB and ENF. For the LLOQ, the criteria for acceptability of the relative standard deviation (RSD) should be less than 20%. For all other concentrations, the RSD has to be less than 15%. 

Carry-over was assessed by injecting a blank sample without IS after injection of the HLOQ containing the two drugs and IS to make sure that there is no impact of carry-over of the method on the accuracy of the study samples. This procedure was carried out six times [19,20]. The detected response should be less than 20% of the LLOQ of each drug and less than 5% of the IS. 

Recovery was calculated for ENF and BNB by comparing responses of extracting plasma samples at three levels (1.5, 1800, and 2400 ng/mL) and blank samples spiked with analyte postextraction at the equal concentrations. The recovery has to be reproducible and consistent over the concentration range. The matrix effect was quantified for ENF and BNB and the IS from six different blank plasma batches. After precipitation with acetonitrile, samples were spiked with the IS and the two analytes at three concentrations—LQC, MQC and HQC. 

The matrix effect (ME) was intended by the ratio of the peak area in the presence (blank spiked with analytes after extraction) and absence of the matrix (pure analytes solution) [21]. IS normalized ME is the ratio of the ME of the analytes to the ME of IS and had to be within 15% of RSD. Six replicates of plasma samples spiked with the HLOQ for ENF and BNB (4500 ng/mL) were diluted two and four times with blank plasma. The resulting concentration was compared to the nominal concentration to acquire if dilution affects accuracy and precision or not. The mean analyzed value should be within 15% of the nominal, and the precision of the replicates should be equal to or less than 15% RSD. 

The stability of ENF and BNB was assessed after exposing the QC samples at LQC, MQC, and HQC to different storage conditions (temperature and time). The applied conditions include short-term stability at room temperature for 24 h in an autosampler for 24 h at 10 ºC. Long-term stability was assessed after storing QCs for 30 days at −80 °C. Freeze and thaw stability were evaluated after three freezing and thawing cycles and compared with freshly prepared QCs. Moreover, the stock solution stability for ENF and BNB and IS was studied at a temperature of 5 ± 2 °C. All sample accuracies should be ± 15% to be considered stable.

### 3.6. Pharmacokinetic Study

Four healthy male Wistar rats (220–250 g) were brought from the Animal Care Center of King Saud University (Saudia Arabia). All experimental procedures were reviewed in accordance with the guidelines of King Saud University Institutional Research Ethics Committee (REC) with an ethics reference number (SE-19–109). Rats were acclimatized for 7 days to laboratory environments before the experiment was directed. Diet was prohibited for 12 h before the experiment, but the water was freely available. The dose of drugs was determined according to the body surface area employing the below-mentioned formula [22].
Human dose (mg/kg) = Animal dose (mg/kg) × animal K_m_/Human K_m_

K_m_ is a factor used in dose conversion. It is a ratio of average body weight (kg) and body surface area (m^2^). K_m_ values for the rat and human are 6 and 37, respectively. Human dose for BNB and ENF was considered as 90 and 450 mg, respectively [15,18]. 

Blood samples (300 μL) were collected into heparinized 1.5 mL polythene tubes containing ethylenediamine tetraacetic acid dipotassium (EDTA K_2_) (anticoagulant) before drug administration and at 0.15, 0.5, 1, 2, 4, 6, 8, 12, 18, and 24 h after oral administration of binimetinib (3.8 mg/kg) and encorafenib (20 mg/kg) [23]. In the current study, both drugs were dissolved in 1% DMSO/saline. The samples were directly centrifuged at 3500 rpm for 10 min at 4 °C. The plasma obtained was stored at −80 °C until analysis. The same method of extraction described under calibration standards preparation (2.4.) was used for sample preparation. The PK parameters of BNB and ENF such as C_max_, T_max_, t_1/2kel_, AUC_0–24_, and AUC_0–∞_ were calculated by fitting the data to a non-compartmental analysis (NCA) model with PK Solver Add-In software [23].

## 4. Conclusions

A newly developed and fully validated LC-MS/MS bioanalytical assay was used to analyze ENF and BNB in rat plasma. The developed wide range of calibration curves of the proposed assay allowed efficient quantitation of pharmacokinetic parameters after oral administration of binimetinib (3.8 mg/kg) and encorafenib (20 mg/kg)**.** The present approach is distinguished by appropriate extraction recovery with the lack of matrix interference. Results also confirmed the high sensitivity of the developed method as low as 0.2 ng/mL with a total run time of 2 min, which rendered the developed assay applicable for effective routine assays in pharmacokinetic studies.

## Figures and Tables

**Figure 1 molecules-28-00079-f001:**
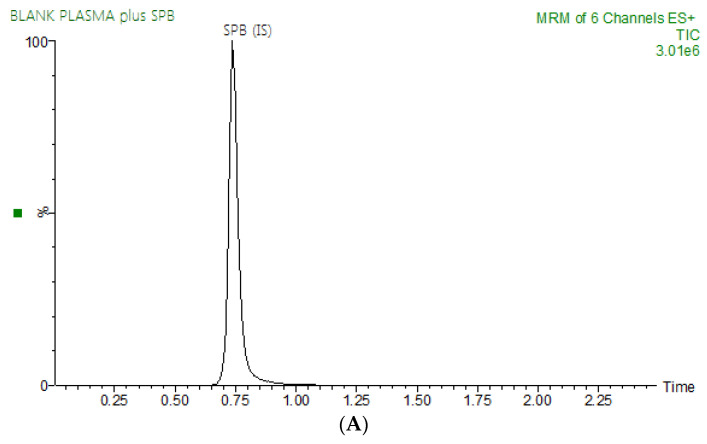
Representative total ion chromatograms for blank rat plasma spiked with spebrutinib (IS) at a concentration of 100 ng/mL (**A**) and overlays of the LC–MS/MS analysis of binimetinib (1.14 min), encorafenib (1.86 min) at concentrations of 0.5–3000 ng/mL and IS (0.73 min) at a concentration of 100 ng/mL (**B**).

**Figure 2 molecules-28-00079-f002:**
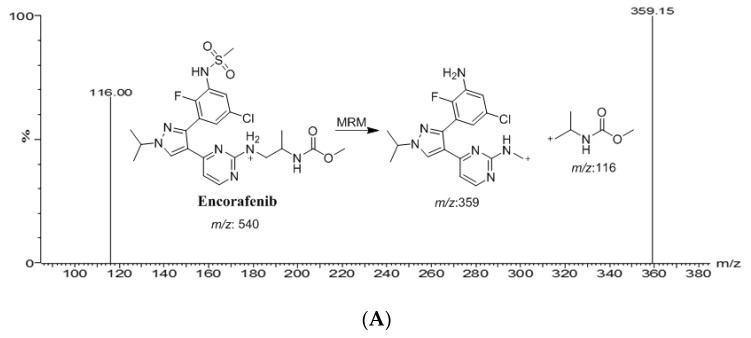
Multiple reaction monitoring (MRM) mass spectra and the expected fragmentation pathway of encorafenib (**A**), binimetinib (**B**), and spebrutinib (**C**) (IS).

**Figure 3 molecules-28-00079-f003:**
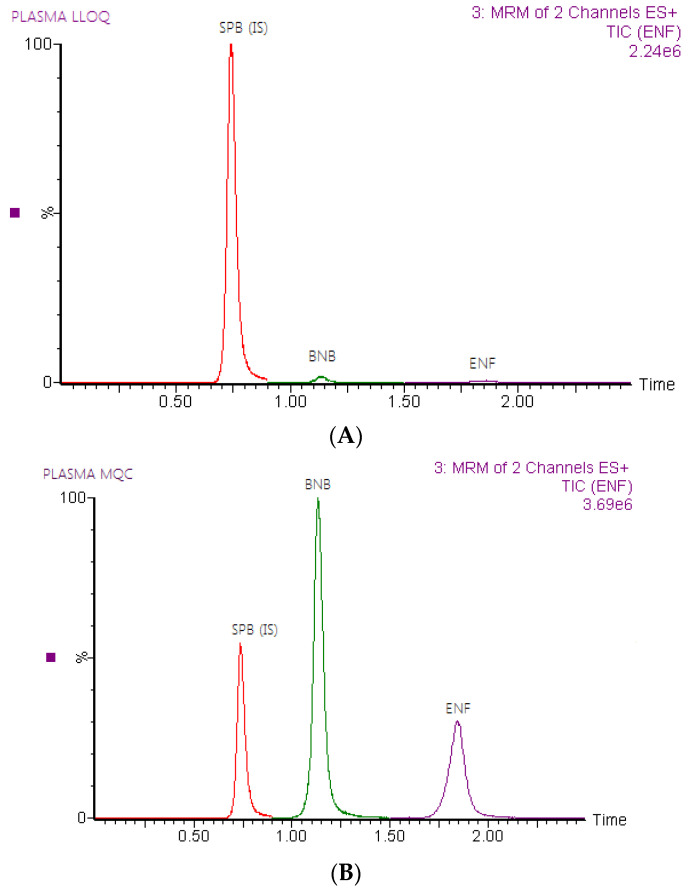
Representative total ion chromatograms of rat plasma spiked with LLOQ (**A**), MQC (**B**), and HQC (**C**); for encorafenib (1.86 min), binimetinib (1.14 min) and IS (0.73 min).

**Figure 4 molecules-28-00079-f004:**
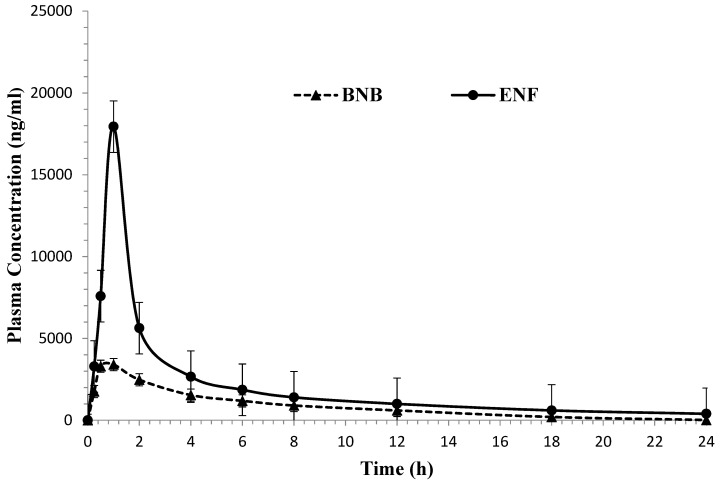
Mean plasma concentration–time profile of binimetinib and encorafenib in rats after a single oral dose of 3.8 mg/kg binimetinib and 20 mg/kg encorafenib (n = 6, mean ± SD).

**Table 1 molecules-28-00079-t001:** LC-MS/MS optimized parameters for the determination of encorafenib, binimetinib and IS.

Drug	Ion Mode	Precursor (*m*/*z*)	Quantification traces (*m*/*z*)	Qualification traces (*m*/*z*)	Cone Voltage (V)	Collision energy (CE, eV)
BNB	+ve	441.0	165.0	149.9	44	54/32
ENF	+ve	540.1	359.1	116.0	54	46/44
IS	+ve	424.1	370.1	58.9	58	32/26

**Table 2 molecules-28-00079-t002:** Statistical parameters of calibration curves for ENF and BNB in rat plasma using the developed LC–MS/MS method.

Parameters	BNB	ENF
Concentration range (ng/mL)	0.5–3000	0.5–3000
Intercept (a)	4.31 × 10^−2^	6.45 × 10^−2^
Slope (b)	2.36 × 10^−3^	1.17 × 10^−3^
Coefficient of determination (r^2^)	0.998	0.999
S_Y/N_ ^a^	7.70 × 10^−3^	6.48 × 10^−3^
S_a_ ^b^	2.42 × 10^−3^	2.04 × 10^−3^
S_b_ ^c^	2.08 × 10^−4^	1.75 × 10^−4^
LLOQ (ng/mL)	0.5	0.5
LLOD (ng/mL	0.2	0.2

^a^ SD of the residual; ^b^ SD of the intercept; ^c^ SD of the slope.

**Table 3 molecules-28-00079-t003:** The accuracy and precision data for the determination of binimetinib and encorafenib in rat plasma.

Analyte	Concentration ng/mL		Inter-Day		Intra-Day	
			RSD (%)	Accuracy (%)	RSD (%)	Accuracy (%)
Binimetinib	LLOQ	0.5	94.00	7.52	95.71	5.17
	LQC	1.5	95.61	2.39	96.45	3.24
	MQC	1800	97.53	1.53	99.18	0.38
	HQC	2400	101.31	0.62	100.31	0.58
Encorafenib	LLOQ	0.5	95.15	6.23	92.88	5.67
	LQC	1.5	97.51	2.62	96.65	2.61
	MQC	1800	96.65	1.57	99.57	0.33
	HQC	2400	99.84	0.97	102.28	1.26
n	6			18		

**Table 4 molecules-28-00079-t004:** Extraction recovery for the analysis of binimetinib and encorafenib and IS in rat plasma by the developed LC-MS/MS method.

Nominal Concentration (ng/mL)	Binimetinib			Encorafenib			IS
	1.5	1800	2400	1.5	1800	2400	100
Mean ^a^	1.39	1706.77	2220.74	1.40	1735.89	2216.03	95.28
RSD	1.09	0.28	2.07	1.08	0.95	1.10	1.74
Recovery (%)	92.33	94.82	92.53	93.33	96.43	92.33	95.28
Mean recovery (%)	93.41			94.18			95.28

^a^ Average of six determinations.

**Table 5 molecules-28-00079-t005:** Evaluation of the dilution integrity of binimetinib and encorafenib in rat plasma.

Analyte	Spiked Conc. (ng/ mL)	Dilution Fold	Mean Recovery (%) ± RSD ^a^
Binimetinib	4500	1:2	94.66 ± 1.13
		1:4	96.86 ± 1.08
Encorafenib	4500	1:2	99.26 ± 1.82
		1:4	98.71 ± 1.10

^a^ Mean recovery (%) ± RSD of six determinations.

**Table 6 molecules-28-00079-t006:** Stability results for binimetinib and encorafenib in plasma at different conditions.

Analyte	Concentration ng/mL		Short Term Stability at Room Temperature (24 h)		Autosampler Stabilityat 10 °C (24 h)		Freeze and Thaw Stability at −80 °C (3 Cycles)		Long Term Stability at −80 °C (30 Days)	
			Recovery (%)	RSD (%)	Recovery (%)	RSD (%)	Recovery (%)	RSD (%)	Recovery (%)	RSD (%)
Binimetinib	LQC	1.5	96.23	2.57	97.35	4.79	104.33	2.45	95.54	2.92
	MQC	1800	103.14	1.75	104.26	2.29	103.67	1.93	103.92	2.26
	HQC	2400	93.39	2.92	95.41	2.24	95.21	2.97	94.76	3.21
Encorafenib	LQC	1.5	95.25	2.89	96.57	2.46	99.47	0.83	97.47	1.71
	MQC	1800	102.64	1.64	99.83	2.55	96.78	2.61	102.24	2.24
	HQC	2400	94.48	3.53	92.44	5.15	96.53	2.12	94.62	3.65
n			3		3		3		3	

**Table 7 molecules-28-00079-t007:** The pharmacokinetic parameters of binimetinib and encorafenib in rat plasma after oral administration of 3.8 mg/kg binimetinib and 20 mg/kg encorafenib (n = 6, mean ± SD).

Parameters	Unit	Binimetinib *	Encorafenib *
AUC_0-t_ ^a^	µg/mL·h	12.35 ± 1.86	30.30 ± 3.01
AUC_0-∞_ ^b^	µg/mL·h	18.16 ± 1.31	36.52 ± 3.92
C_max_ ^c^	µg/mL	3.43 ± 0.46	16.42 ± 1.47
T_max_ ^d^	h	1	1
Cl/F ^e^	L/h	0.21 ± 0.03	0.55 ± 0.06
t_1/2kel_ ^f^	h	3.39 ± 0.43	2.48 ± 0.24
MRT_0-∞_ ^g^	h	4.50 ± 0.28	3.23 ± 0.40

* Data are presented as mean ± SD; ^a^ area under the curve up to the last sampling time; ^b^ area under the curve extrapolated to infinity; ^c^ maximum plasma concentration; ^d^ time taken to reach the maximum plasma concentration; ^e^ total clearance of the drug from plasma after oral administration; ^f^ half-life in elimination phase; ^g^ mean residence time.

## Data Availability

Not applicable.

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
