# Peer review of "A Rapid and Sensitive Liquid Chromatography-Tandem Mass Spectrometry Bioanalytical Method for the Quantification of Encorafenib and Binimetinib as a First-Line Treatment for Advanced (Unresectable or Metastatic) Melanoma—Application to a Pharmacokinetic Study"

_molecules, 2022, doi:10.3390/molecules28010079_

Round 1
Reviewer 1 Report
Manuscript Number: Molecules-2063042
Manuscript title: A Rapid and Sensitive Liquid Chromatography-Tandem Mass Spectrometry Bioanalytical Method for the Quantification of Encorafenib (BRAF Inhibitor) and Binimetinib (MEK Inhibitor) as a First-Line Treatment for Advanced (Unresectable Or Meta-static) Melanoma. Application to a Pharmacokinetic Study
A) Major
- There is little originality in this research – there is similar method and identical drugs in the paper [1]. The only differences in present manuscript compared to mentioned article [1] are:
· different mass spec and hplc producer
· a little different chromatographic conditions
· different matrix
Because of only few differences between this two manuscripts in reviewer opinion presented work is more like optimization of method described in [1] rather than development of method. Nevertheless from technical point o view the experiments were well conducted and results were described properly. Additionally the Authors properly cited mentioned manuscript.
B) Minor – there are a few of inaccuracies in presented manuscript:
- The title is too long – maybe the Authors should remove information in brackets.
- In the abstract there is lack of PK study results. The Authors may add the result of PK study, instead of first few sentences which in reviewer opinion are unnecessary.
- There is lack of the meaning of the abbreviations in abstract section.
- The introduction is focused in 70% in drugs description and only in 30% in aim of the conducted study. If the aim was to developed method and its used in PK study, there should be opposite proportions.
- The meaning of the abbreviations in main manuscript are later than used of abbreviations.
- In Result & Discussion section there is lack of meaning of the IS abbreviation.
- Line 170 please add the meanings of quality control points abbreviations.
- The matrix effect is more proper than matrix factor – line 188.
- Figure 4 is unnecessary.
- The Authors should shortly describe the PK protocol in M&M section – which model was used, what parameters were calculated.
- In table 7 the Authors used t1/2 abbreviation – it should be precise which half-life it is – probably t1/2β = t1/2kel which is half-life in elimination phase.
- Line 261 – there is probably mistake of the name of equipment names and producer – in reviewer opinion Acquity UPLC is produced by WATERS (is the UPH the same as UPLC H-class system?) and the mass spec is Xevo TQD not the Acquity TQD.
- There is a few extra space between words – lines: 25, 68, 160, 169, 172, 182, 184, 204, 233, 357.
References
[1] Attwa, M.W.; Darwish, H.H.; Al-Shakliah, N.S.; Kadi, A.A. A Validated LC–MS/MS Assay for the Simultaneous Quantification of the FDA-Approved Anticancer Mixture (Encorafenib and Binimetinib): Metabolic Stability Estimation. Molecules. 2021, 26, 2717.
Reviewer 2 Report
This is an interesting paper and is well designed and carried out. It is difficult to achieve the fast, high sensitivity and selectivity the authors did while using a small sample volume, in a complex matrix. There are some minor typographical errors like in L 277-8 where "H" should be "h" (for hour?) and L 303 should have "vortexed' rather than "vortexes'. The only improvement I can suggest is for a comparison of m/z ion ratios obtained for the sample and standard solutions as further confirmation of peak purity and identity.
Round 2
Reviewer 1 Report
.